# Stroboscopic operando spectroscopy of the dynamics in heterogeneous catalysis by event-averaging

Jan Knudsen [1,2 ✉], Tamires Gallo[1], Virgínia Boix [1], Marie Døvre Strømsheim [3], Giulio D'Acunto [1], Christopher Goodwin [4], Harald Wallander [1], Suyun Zhu[2], Markus Soldemo [4], Patrick Lömker [5], Filippo Cavalca [2], Mattia Scardamaglia[2], David Degerman [4], Anders Nilsson[4], Peter Amann [4], Andrey Shavorskiy [2] & Joachim Schnadt [1,2]

Heterogeneous catalyst surfaces are dynamic entities that respond rapidly to changes in their local gas environment, and the dynamics of the response is a decisive factor for the catalysts' action and activity. Few probes are able to map catalyst structure and local gas environment simultaneously under reaction conditions at the timescales of the dynamic changes. Here we use the CO oxidation reaction and a Pd(100) model catalyst to demonstrate how such studies can be performed by time-resolved ambient pressure photoelectron spectroscopy. Central elements of the method are cyclic gas pulsing and software-based event-averaging by image recognition of spectral features. A key finding is that at 3.2 mbar total pressure a metallic, predominantly CO-covered metallic surface turns highly active for a few seconds once the $O_2$:CO ratio becomes high enough to lift the CO poisoning effect before mass transport limitations triggers formation of a $\sqrt{5}$ oxide.

[1] Division of Synchrotron Radiation Research, Department of Physics, Lund University, Lund, Sweden. [2] MAX IV Laboratory, Lund University, Lund, Sweden. [3] Department of Chemical Engineering, Norwegian University of Science and Technology (NTNU), Trondheim, Norway. [4] Department of Physics, Stockholm University, Stockholm, Sweden. [5] Deutsches Elektronen-Synchrotron DESY, Hamburg, Germany. ✉email: jan.knudsen@sljus.lu.se

One of the key questions in heterogeneous catalysis is how the catalyst structure, composition, and function relate to each other. The fact that the structure of a catalyst is dynamic rather than static as the reaction is initiated causes a major challenge in the endeavour of understanding catalysis. The dynamics are strongly influenced by the exact reaction conditions in terms of temperature, pressure, and gas composition[1,2]. These conditions may fluctuate with time, which means that it is necessary to diligently control and map the conditions at the site of the catalyst. The derivation of the complete chemical mechanism of a particular catalyst cannot be attained without a profound understanding of the dynamics of the catalyst material, irrespective of whether e.g., an automotive three-way catalyst or a catalyst for power-to-X technologies[3] is considered. The fluctuations can be produced by external stimuli, but also by changes being induced by the action of the catalyst itself[4,5].

Changes in the local gas composition above the catalytically active surface[6] and its influence on the surface structure and activity of the catalyst are particularly difficult to address experimentally. The reason is that this requires experimental methods that simultaneously can probe the surface structure, catalytic activity, and local gas composition with a time resolution comparable to the timescale of the structural changes. While picosecond time resolution is necessary if the elemental steps of a catalytic cycle are to be probed[7], millisecond to second time resolution is sufficient to follow structural surface changes of the catalyst, such as phase transitions, surface roughening and segregation. Therefore, methods that allow video-rate frequencies or slightly beyond render the possibility to follow the dynamic restructuring of the catalyst surface. Imaging techniques, such as scanning tunnelling microscopy, transmission electron microscopy, or low energy electron microscopy, can provide time-lapsed surface images at video-rate frequencies in both ultrahigh vacuum and ambient conditions. However, most of these techniques lack the ability to probe the local gas composition with a notable exception being energy-loss spectroscopy in operando TEM experiments[8]. Unfortunately, this technique has limited flexibility for the sample and its environment. Furthermore, surface characterization can be challenging at elevated temperatures and most imaging techniques also lack direct chemical sensitivity. In contrast, ambient pressure x-ray photoelectron spectroscopy (APXPS)[9] is a chemically sensitive technique and it probes both the surface structure and its local gas composition, which gives direct information on the catalytic activity. Moreover, APXPS is rather flexible in terms of the sample environment. However, the overall probability of photoelectron creation and detection is low, which implies that typical spectral acquisition times are relatively long, in the range of several seconds to minutes. Video-rate frequency mapping, with the recording of several spectra per second, occurs at a low signal-to-noise ratio in APXPS. This significantly hinders a proper analysis of the time-resolved APXPS data.

Here, we demonstrate how the fundamental problem of a low signal-to-noise ratio in time-resolved APXPS can be overcome by *event-averaging*. Here, *event-averaging* refers to measuring many repeated times the spectroscopic signal of a certain event and subsequently averaging the signal. The event needs to be always identical—here it is a surface phase transition—and it may or may not be periodic. We use a sample compartment with a small gas volume to expose the sample surface to a rapidly oscillating gas composition with a repetition period between 14 and 145 s. In this way, we force the catalytic surface to periodically switch back and forth between phases with different activities as the catalytic reaction proceeds. The electron energy analyzer in the APXPS system is used to simultaneously and continuously monitor the surface chemical reaction, phase transitions, and gas-phase

composition at framerates of 6−17 Hz. Image recognition and lock-in techniques are applied to the raw data to form an event-averaged signal of many phase change events on the surface. Ultimately, we demonstrate that one can record spectra with an excellent signal-to-noise ratio at a time resolution down to 60 ms.

As a model system, we use the CO oxidation reaction over the Pd(100) surface prototypical for new experimental in situ tools for catalysis. Examples of this include planar laser-induced fluorescence (PLIF)[10], high-energy surface x-ray diffraction (HESXRD)[11], high-pressure scanning tunnelling microscopy (HPSTM)[12], and APXPS[13]. The reaction is also of fundamental interest since chemisorbed structures, thin surface oxides, and bulk oxides all can form at reaction conditions. Which surface is most active has been debated for decades, cf. the recent review by van Spronsen et al.[14]. Thus, in situ HPSTM[15], SXRD[16], APXPS[13], and HESXRD[17] studies in combination mass spectroscopy have shown that the formation of various oxides on the Pd(100) surfaces coincides with the rapid increase in the $CO_2$ production. This observation has been interpreted as evidence for the oxides being more active than the metallic surface. In contrast, older studies by Gao et al. that used polarization-modulation infrared reflection absorption spectroscopy (PM-IRAS) ([18] and references in [14]) found that the $CO_2$ production decreased upon oxidation of the Pd(100) surface. Gao et al. also pointed out that the highly reactive metallic surface state often is transient in high-pressure reactions—and thus only exists for a short time—and that it therefore easily can be overlooked. A recent study by Gustafson et al. carefully compared oxidized and chemisorbed phases and concluded that thin Pd oxide films are at least as active as the metallic surface, while thicker oxides are less active[19]. Other recent temperature-programmed reaction spectroscopy studies by Weaver et al.[20] and Mehar et al.[21] on Pd(111) and Pd(100) compared the intrinsic CO oxidation activity at UHV conditions and concluded that the chemisorbed oxygen phase is two to three times more active than a multilayer PdO(101) film supported on Pd(111)[20], while a multilayer PdO(101) film is more active than a single-layer PdO(101) film grown on Pd(100)[21]. Dynamic measurements with a time resolution comparable to the timescale of the phase change on the surface would provide a powerful method to investigate the transition between chemisorbed and oxidized phases and can potentially be used to identify transient phases[18].

Here, we study the oxidation of CO over Pd(100) at 3.2 and 100 mbar total pressure in two different sets of experiments. Gas pulses are used to drive the surface into different states, and time-resolved, event-averaged APXPS data are obtained and analyzed. The key finding is that the local CO and $O_2$ partial pressures just above the surface trigger surface restructuring. Specifically, we demonstrate that a transient metallic and partly CO-covered surface is highly active for CO oxidation and that it exists for a few seconds before the local gas composition forces the formation of a thin surface oxide. Finally, we discuss how the combination of gas pulsing and event-averaging of data can be applied to other surface science techniques used in catalysis research.

## Results

**Event-averaging by image recognition: 3.2 mbar CO oxidation.** In the first set of experiments performed at the HIPPIE beamline[22] at the MAX IV Laboratory we exposed a Pd(100) surface, located in a cell volume of ∼ 1 l to 58 alternating pulses of CO-rich and $O_2$-rich gas mixtures with CO:$O_2$ mixing ratios of 2.7:1 and 1:2.7, respectively. The total pressure and total flow were kept constant at 3.2 mbar and 9.6 sccm, respectively, while the sample temperature only was semi-constant at 550 ± 10 K, due to the exothermic heat and the reaction that turned on and

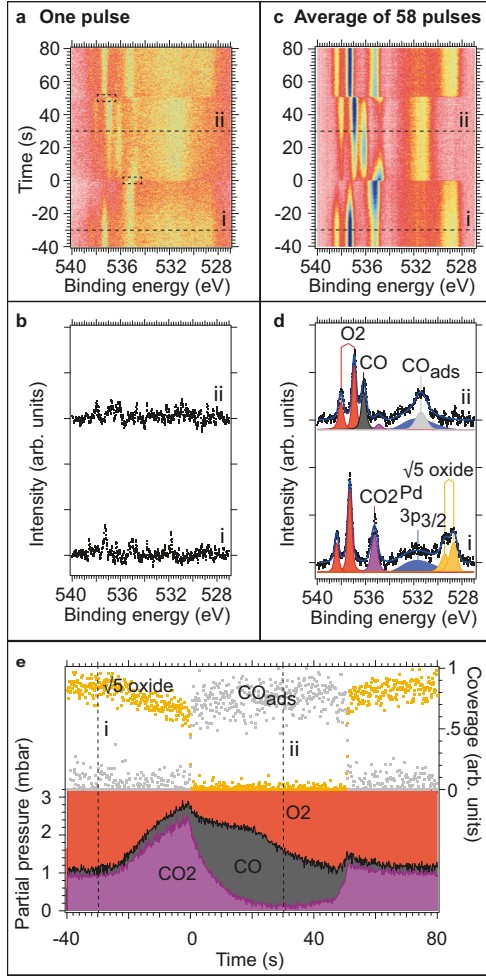

**Fig. 1 Analysis of time-resolved O 1s data recorded at 3.2 mbar before and after event-averaging. a**, **c** Image plots of time-resolved O 1s data acquired with a photon energy of 650 eV before and after event-averaging 58 pulses. **b**, **d** Single spectra recorded along the dotted lines i and ii in panels (**a** and **c**) and the curve fitting and peak assignment for the event-averaged data. **e** Partial pressures, $\sqrt{5}$ oxide, and CO coverage as obtained by curve fitting all spectra in panel (**c**). Partial pressures are shown on vertically stacked y-axes—i.e., at $t = -40$ s the partial pressure of CO is zero. The following colours and markers are used in panels (**d**, **e**): $O_2(g)$ red, $CO(g)$ grey, $CO_2(g)$ purple, Pd $3p_{3/2}$ blue, $CO_{ads}$ light grey squares, and $\sqrt{5}$ oxide intensity orange squares.

off. The duration of the CO-rich pulse was 45 s and that of the $O_2$-rich one 100 s, resulting in a total pulse period of 145 s (further details in the "Method" section). By exposure to these periodically varying pulses, the surface was brought to oscillate back and forth between an inactive and active surface state for the CO oxidation reaction, as verified by mass spectrometry that probed the gas entering the nozzle of the electron analyzer (see Supplementary Fig. 2)

Figure 1a shows an O 1s image plot of 810 spectra (120 s acquisition time) acquired during one complete surface oscillation cycle. The spectral features originate both from the gas phase molecules probed in the volume defined by the photon beam above the sample surface (strong features above 535 eV) and from the surface atoms and adsorbed molecules (weak features below 533 eV). The spectra marked by two dotted lines i and ii in Fig. 1a are shown in Fig. 1b. Clearly, a poor signal-to-noise ratio makes peak decomposition in these spectra impossible. The standard solution for further component analysis would normally be to

time bin the dataset at the expense of time resolution. Information about how fast the phase change occurs on the surface, on hidden transient phases, and on how the catalytic properties change during the phase change would get lost. We note that the experiments were performed at an APXPS beamline with one of the most intense photon fluxes in the world, which renders further improvement of the signal-to-noise ratio by choice of photon source unrealistic at the present stage. In the following, we will show that calculation of the event-averaged signal of the experimental data from all 58 gas pulses instead of the time-averaged signal of a single pulse provides a suitable pathway towards a much-improved signal-to-noise ratio. This then makes possible a detailed component analysis without loss of time resolution.

When event-averaging is employed, it is important to define the event used. Possible event signals could for example be (i) the switching event when the feed-gas composition is changed, (ii) a particular change in the gas composition above the catalyst surface, or (iii) a structural change on the catalyst surface.

Using the switching event when the feed-gas composition is changed event-averaging for a certain delay time ($\Delta t$) can then be achieved by repeated injections of CO-rich pulses ($t_0$ denotes the injection time of the CO-rich pulse) and summation of many spectra recorded after a fixed delay time ($\Delta t$). By repeating such a conventional *pulse-probe* experiment for many different time delays a detailed understanding of the time evolution can potentially be obtained. However, in the present case, the mass flow controllers—switching the gas composition—are located far away from the surface. For example, it took ~120 s from the gas switching event before the gas composition started to change in the measurement cell. Moreover, the ~1 l cell volume leads to a slow, rather than a direct change of gas composition in the cell volume. This implies that the gas pulses were not completely reproducible in terms of their arrival time, and the gas-composition oscillation in the cell was therefore not strictly periodic. This is further illustrated in Supplementary Fig. 5. In the same way, segregation of impurity atoms from the bulk of the single crystal to the surface or slow contamination of the surface by residual gas impurities may generate a response of the catalyst to the oscillating gas composition that fails to be strictly periodic. Therefore, the gas switching event is far from an ideal event signal to average over. In particular, one should be careful with doing this on a general purpose APXPS setup or any other in situ experimental setups not optimized for pulse-probe experiments.

Rather than using the gas switching event on a general-purpose APXPS setup without strictly periodic pulses, we here will use an internal event signal found in the measured APXPS spectra which signals a structural change on the surface. Appropriate locking signals are expected to be found at the beginning and the end of the CO pulse as seen by the sample in the cell. These two events need to be considered separately since the lengths of the different parts of the gas pulse vary slightly over time, which shows that the gas pulses fail to be strictly periodic as already discussed above (cf. Supplementary Fig. 5). Any feature in the image plot of Fig. 1a that signals a phase transition on the surface can potentially be used as the locking signal, as the time-resolved data were recorded continuously. Examples of such features in the raw data can be the appearance or disappearance of a signal or the changed intensity of a component due to adsorption of molecules or oxide formation. It can also be a work-function shift on the surface, which is visible as an apparent binding energy shift of the gas phase components[23]. By trial-and-error of different locking signals, we found that the work function-induced shift of the gas phase O 1s peak of $CO_2$ (start of CO pulse) and $O_2$ (end of CO pulse) is the most suitable locking signal. These shifts are visible as clear image contrasts within the dotted rectangles in

Fig. 1a, and one stamp-image within each rectangle is chosen. An image-recognition algorithm loops over the raw data and searches for the best match between these stamp signals in all of the data (see Supplementary Movie 1). Hereby, we can determine the absolute time of each phase transition event on the surface with a time resolution that corresponds to the sampling frequency of the experiment, giving 148 ms time resolution in the present experiment. Hence, the data from all 58 pulses can be properly time-aligned and the event-averaged signals at the start and end of the CO-pulse can be constructed by averaging the time-aligned data of all 58 pulses. Finally, the two event-averaged image signals are merged together to form the full event-averaged image (Fig. 1c).

Clearly, the averaging algorithm has resulted in a much-improved signal-to-noise ratio without any loss in time resolution. This is already clear from a consideration of the image plot in Fig. 1c and even clearer from the appearance of the two spectra i and ii in Fig. 1d extracted from Fig. 1c as indicated. Deconvolution of these spectra is now possible with high precision in contrast to what can be done on the data in Fig. 1b. Similar event-averaged $C\ 1s$ and $Pd\ 3d_{5/2}$ data are shown in Supplementary Figs. 6 and 7.

**Higher repetition rate and pressure: 100 mbar CO oxidation.** We demonstrate the capability of the method with a second dataset of CO oxidation Pd(100) at higher pressure recorded on the POLARIS instrument at DESY (cf. the "Methods" section and the SI)[24]. This setup takes advantage of a small 30 μm aperture of the analyzer and a short 30 μm sample-cone distance, which leads to the creation of a *virtual cell* with μm dimensions. A total flow of 1800 sccm was used and resulted in a local gas pressure of ~100 mbar.

Figure 2 uses a similar layout as Fig. 1 and summarizes the CO oxidation experiments at 100 mbar. Here, the gas pulse length was reduced to 14 s, which was made possible by the very short gas-exchange time. For the measurements, a pulsing pattern of 8 s oxidation in pure $O_2$ and 2 s reduction in 8:1 $CO:O_2$ was used. To avoid continuous mixing of CO and $O_2$ with a composition between the lower and upper explosive limit in the gas inlet line 2 s of 1:8 $O_2$:He was used at each side of the CO-rich pulse. Due to the He pulses causing increased electron transmission through the gas phase we observe increased intensity of the Pd $3p_{3/2}$ line at both sides of the CO rich pulse (seen just above the dotted lines marked with i and ii in Fig. 2a). During the experiments, the crystal was kept at a stable temperature of 803 ± 2 K, as the rapid gas compositions oscillations left a short time window for the crystal to cool or heat due to the absence or presence of exothermic heat from the CO oxidation reaction.

The short gas pulse length allowed us to measure and average over more than 500 pulses within the same measurement time as was used for 58 gas pulses in the first experiment. Since the gas dosing was very reproducible with the high gas flows and since oxidation of a thicker Pd oxide film is a slower process as we will see below only one stamp-signal was used (dotted rectangle in panel a). Instead of using the work-function shift as was done in Fig. 1, here we found that the Pd $3p_{3/2}$ component of the metallic surface was the optimum locking signal.

Clearly, the comparison of the two topmost panels of Figs. 1 and 2 demonstrate how noisy raw data recorded in a transient gas flow by event-averaging can be transformed into high-quality data with a dramatically improved signal-to-noise ratio.

## Discussion
The low-noise $O\ 1s$ data in Figs. 1d and 2d derived using the above method, are well suited for further analysis, since they

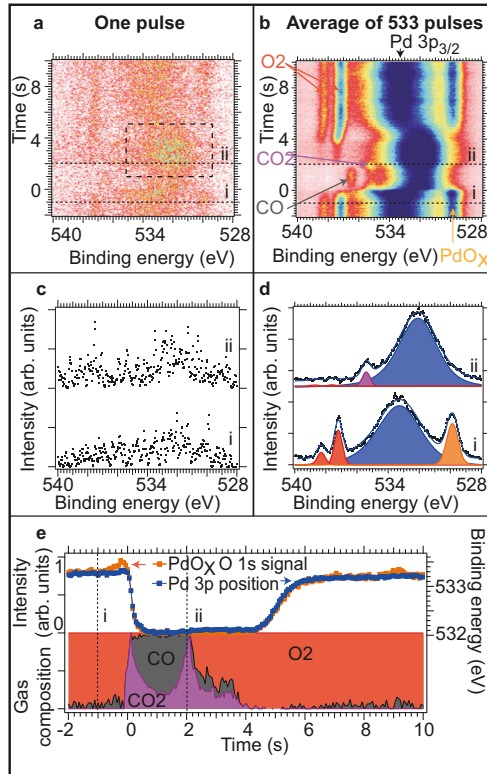

**Fig. 2 Analysis of time-resolved $O\ 1s$ data recorded at 100 mbar before and after event-averaging. a, b** Image plots of time-resolved $O\ 1s$ data acquired with 17 Hz and a photon energy of 4.6 keV before and after event-averaging 533 pulses. **c, d** Single spectra recorded along the dotted lines i and ii in panels (**a, b**) including curve fitting for the event-averaged spectra. Components are defined in panel (**b**). **e** Relative gas composition, PdO$_x$ $O\ 1s$ intensity signal, and binding energy position of Pd $3p_{3/2}$ peak as obtained by curve fitting all spectra in panel (**b**). Relative gas composition is shown on vertically stacked y-axes. The following colours and markers are used in panels (**d, e**): $O_2$(g) red, CO(g) grey, $CO_2$(g) purple, Pd $3p_{3/2}$ binding energy position blue squares, and PdO$_x$ oxide intensity orange squares.

contain simultaneous information on all gas-phase species, the surface-adsorbed CO species, and the oxidation state of the surface. The time-dependent gas composition just above the sample surface can be obtained from the curve fitted intensity of the gas phase components divided by the number of oxygen atoms of each molecule—see Figs. 1e and 2e. Note that in the case of Fig. 1e partial pressures are calculated from the known total pressure and the measured gas compositions, while Fig. 2e displays the relative gas composition as we do not measure the partial pressure of He. Partial pressures, which represent the gas composition, are shown with vertically stacked y-axes to the left in both cases meaning that the partial pressure of CO is zero at $t = -40$ s in Fig. 1e. The entirety of curve fitting results is available from Supplementary Movies 2 and 3.

**Steady-state surface oxidation and reduction at 3.2 and 100 mbar.** The surface components in Fig. 1d are the Pd $3p_{3/2}$ line (blue), the signal in grey at 531.4 eV binding energy due to adsorbed CO, and the two components at 528.7 and 529.3 eV (orange), which we attribute to a $(\sqrt{5} \times \sqrt{5})R27°$ surface oxide (hereafter named $\sqrt{5}$ oxide). The assignment agrees with previous studies[13,25]. The sum of the intensities of the $\sqrt{5}$ oxide components is shown as orange data points in Fig. 1e together with the time-dependent intensity of the surface CO peak. Both curves are normalized to the value of the highest intensity.

In contrast, in Fig. 2e we find a surface $O\ 1s$ component at 529.9 eV due to a thick $PdO_x$ phase, displayed in orange colour, and an intense Pd $3p_{3/2}$ line due to the high kinetic energy of the photoelectrons in this set of experiments. The former signal is present in oxygen-rich conditions only. Its varying intensity correlates well with the position of the $Pd\ 3p_{3/2}$ component, which changes between 533.2 eV in oxygen-rich and 532.0 eV in CO-rich conditions. First of all, this implies that the surface in CO-rich conditions is entirely reduced to its metallic state. Secondly, the palladium oxide phase in $O_2$-rich conditions must be thicker or comparable to the probing depth of approximately 17 Å (see SI for details) since the bulk-metal phase cannot be discerned anymore, as is visible from the observation that the entire $Pd\ 3p_{3/2}$ peak shifts. This conclusion is further supported by the $Pd\ 3d_{5/2}$ data in Supplementary Fig. 8, which suggests a $PdO_X$ thickness of 25 Å.

A comparison of the data in Figs. 1 and 2 make it clear that the high $O_2$ pressures of 100 mbar lead to a much higher degree of oxidation of the Pd surface than in the experiments performed at 3.2 mbar, in which only a surface oxide is formed. Independent of the degree of oxidation the exposure of the surface to CO-rich conditions leads to a complete oxide removal, most clearly demonstrated by the $Pd\ 3d_{5/2}$ data in Supplementary Figs. 6 and 8.

**Time-resolved analysis of the 3.2 mbar experiments**. Regarding the 3.2 mbar experiment (Fig. 1), the observed formation of a √5 oxide on Pd(100) at the present experimental conditions has been verified by several techniques before, including APXPS[13,25]. It is well known that the surface system is in the so-called CO mass-transfer limit in this pressure regime[5,17], with the formation of CO depletion layer. Previously it has, however, not yet been possible to determine the exact dynamics of the catalytic process and in particular what happens in the transition region into and out of the mass-transfer limit. The high time resolution and good signal-to-noise ratio of the data in Fig. 1c allow us to provide such analysis now.

Figure 1e shows that the CO pulse starts to arrive at the surface at around −25 s and that the $CO_2$ production increases. The increased CO content in the pulse presses more and more CO through the depletion layer above the surface. This CO reaches the surface where it reacts instantaneously and forms $CO_2$ within the time resolution of the experiment; thus, the $CO_2$ production follows suit. Simultaneously, as the CO pulse produces more $CO_2$ a slow decrease of the √5 oxide is observed between −25 s to just before 0 s suggesting slow reduction. The curve-fitting of the $Pd\ 3d_{5/2}$ data presented in Supplementary Fig. 6 also suggests slow reduction of the √5 oxide within this time interval. Careful inspection of Fig. 1e also shows that there are small and increasing amounts of CO in the gas phase above the sample surface at the same time as the √5 oxide intensity slowly is decreasing. This suggests that the rate of oxide removal is slightly higher than the replenishment rate in the time interval from −25 to just before 0 s.

Independently of this slow initial removal of the √5 oxide it is clear that its rapid removal first is triggered when only little $O_2$ remains above the surface. Now the system tips over into the $O_2$ mass-transfer limit and the √5 oxide is not replenished anymore. At $t = 0$ s, the supply of O atoms from the √5 oxide to the CO oxidation reaction leads to a fast disappearance of the oxide signal; now, instead CO covers the entire surface—poisons it—and thus stops the reaction to $CO_2$. The switch-over in surface coverage from oxygen to CO is very rapid, so rapid, indeed, that the transition cannot be resolved with the sampling frequency of the experiment corresponding to a time resolution of 148 ms.

These observations show that all CO is converted to $CO_2$ as long as sufficient $O_2$ is available in the gas phase to replenish, or partly replenish, the oxide. Conversely, once this is not the case anymore, the phase transition from the oxide-covered surface to the CO-poisoned one is very rapid.

The CO partial pressure reaches its maximum at $t = 20$ s as expected due to the 45 s pulse duration and the pulse arrival at $t = −25$ s. Hereafter the local CO pressure above the surface starts to decrease due the oxygen-rich gas composition that follows the CO pulse. $CO_2$ production is resumed first at $t = 38.0$ s when the CO partial pressure has dropped to ~1 mbar (at $t = 38$ s). At $t = 47$ s the $CO_2$ pressure starts to increase rapidly and the surface transition into the CO-uninhibited regime. This transition seems to be triggered by a sufficient increase in the gas-phase $O_2$:CO ratio, which presumably lifts the CO poisoning effect and allows oxygen to chemisorb on the metal in sufficient quantities to trigger $CO_2$ production. First at $t = 50.8$ s, when the mass transfer limit of CO is reached the √5 oxide signal reappears, marking the transition from the CO-covered to the √5 oxide-covered surface.

Altogether, our results of the 3.2 mbar experiment show that the surface oxidizes CO to $CO_2$ efficiently once the √5 oxide is present at the surface, and that it maintains the reaction in or near the mass transfer limit in gas mixtures ranging from oxygen-rich and all the way to nearly stoichiometric conditions. As the reaction is kept in or near the mass transfer limit no or little CO is found in the gas phase near the surface and a reduction of the √5 oxide is impossible or slow. However, once the oxygen supply from the gas phase starts to become the limiting factor the √5 oxide is reduced rapidly and within less than 148 ms. The fact that the reduction proceeds rapidly explains also why no CO poisoning is observed when the intensity of √5 oxide slowly decreases (−25 to 0 s). Here, the √5 oxide serves as an oxygen reservoir that rapidly can release oxygen if the concentration of chemisorbed oxygen should start to decrease anywhere on the surface. In this way CO poisoning is prohibited until the oxygen reservoir is emptied. Similarly, the reverse transition from a CO- to the √5 oxide-covered surface happens first when the surface system reaches the CO mass-transfer limit. In lack of CO supply, the oxygen coverage increases on the surface and the oxygen reservoir in the form of the √5 oxide is filled rapidly again within less than 148 ms. While the surface always is active and in the MTL once the √5 oxide is present, the metallic and mainly CO-covered surface may be very active, but only if the $O_2$:CO ratio is sufficiently high to lift the CO poisoning effect.

The probing of the surface and the local gas composition right above the surface with sub-second time resolution is essential for drawing all these conclusions. Previously, PLIF, for local gas-phase probing, and HESXRD[26] or surface reflectance[19], for surface analysis, have been synchronized in order to obtain similar information. Such a combination has the advantage that structural information is accessible directly, but it lacks the chemical sensitivity of APXPS. It is, however, interesting to note the similarity of the present results to those of Blomberg et al.[26], who exposed a Pd(100) crystal to a gas mixture of 6 mbar CO and 24 mbar $O_2$ at a sample temperature of 473 K. They found by PLIF that $CO_2$ production reached saturation approximately 2.5 s before any sign of the √5 oxide was observed by HESXRD. The finding matches well with the present results. The fact that a high $O_2$:CO ratio is required before the √5 oxide develops is also well in line with a recent study by Gustafson et al.[19], who also concluded that the √5 oxide is at least as active for CO oxidation as the metallic surface. While the study by Gustafson et al. concluded that the surface oxide must be as active as the metallic surface and the study by Blomberg et al. concluded that the surface is not fully oxidized when the ignition of the catalyst

occurs, the chemisorbed phase present on the metallic surface when ignition occurs was left untouched. The reason for this is clear. These previous studies did not have the ability to probe the oxide, chemisorbed phases, and localized gas composition simultaneously with high time resolution. As pointed out by Gao et al.[18] more than 10 years ago the complexity of high-pressure reactions, and especially the limitations of mass transfer, results in high reaction rate states which often are transient. Here, we show that this indeed is the case, and surprisingly the majority of the surface is CO-covered according to Fig. 1e when ignition occurs. First, when the mass transfer limit is reached and no CO is found near the surface all CO desorbs and the oxygen reservoir in the form of the surface oxide is filled quickly.

The observation that the mass transfer limit is reached before the surface oxide is formed is a key finding of our work and resolve the dilemma of the chicken and the egg: Is the mass transfer limit reached because the √5 oxide appears or is the √5 oxide formed because the mass transfer limit is reached? We clearly show that the latter scenario is the case and even provide clear evidence that it is a partly CO-covered and metallic surface that is responsible for bringing the system back into the mass transfer limit.

**Time-resolved analysis of the 100 mbar experiments**. Turning to the results of the 100 mbar-experiment summarized in Fig. 2, we find that the front of the arriving CO-rich pulse of 2 s duration is entirely consumed by the CO oxidation reaction (Fig. 2e): a $CO_2$ cloud above the surface builds up between $t = 0.0$ to $t = 0.1$ s, and the gas composition above the surface consists of pure $CO_2$. The complete conversion of CO is accompanied by a reduction of the oxide, which is fully consumed within ∼ 250 ms. The surface becomes metallic but remains active. All $O_2$ that reaches the surface reacts with CO to form $CO_2$—i.e., now the reaction is limited by the oxygen supply, since a 8:1 $CO:O_2$ ratio is used for the CO-rich pulse. The tailing-off of the CO-rich pulse goes along with a second increase of the $CO_2$ production at 2 s. We attribute this second phase of high $CO_2$ production rate to the gas mixture becoming stoichiometric. Clearly, at this point, no $PdO_x$ surface phase is observed; the metallic Pd surface is sufficiently active for the complete combustion of all incoming CO. After the second peak in $CO_2$ production the reaction becomes CO mass-transfer limited, and therefore more and more $O_2$ is observed in the gas above the surface.

The overall picture that evolves after inspection of Fig. 2e is that the catalytic reaction is mass-transfer limited during its entire time span. First, the reaction is CO mass transport limited, then $O_2$ limited, and finally CO limited again. The clearest evidence for this is that significant amounts of CO and $O_2$ never are observed at the same time. At each transition between the CO and $O_2$ mass-transfer limits the gas composition becomes stoichiometric and a high intensity of pure $CO_2$ in the gas phase is observed. Therefore, the situation is very different from Fig. 1 where a sufficient increase in the gas-phase $O_2:CO$ ratio lifted the CO poisoning effect and allowed oxygen to chemisorb on the metal in sufficient quantities to trigger $CO_2$ production. One direct consequence of the mass transfer limitations in Fig. 2 is that the $PdO_x$ reduction speed most likely is dictated by the limited CO delivery to the surface. Therefore, it is impossible to measure the intrinsic reactivity of the $PdO_x$. Nevertheless, it is clear that the re-oxidation of the surface to $PdO_x$ at the end of the CO-rich pulse is much slower than the reduction of the oxide upon arrival of the pulse (cf. a further analysis of the event-averaged $Pd\ 3d_{5/2}$ data in Supplementary Fig. 8 and its comparison to the $PdO_x$ signal as determined from the $O\ 1s$ and $Pd\ 3d$ signals).

The finding that $CO_2$ production is observed when a thick $PdO_x$ oxide is found on the surface matches well with a recent SXRD study of Shipilin et al.[28]. In this study, different Pd oxide phases were observed with SXRD while in the mass transfer limit for the CO oxidation reaction. In low excess of oxygen in the feedgas the authors observed growth of PdO islands atop a √5 surface oxide, while a thick and polycrystalline PdO film were observed with a high excess of oxygen in the feedgas. While the study of Shipilin et al. used the composition of the feedgas to tune the gas atmosphere that the surface equilibrates to, we measure the gas composition locally and clearly show that high pressure of pure oxygen is a requirement for the development of the thicker $PdO_x$ oxide. In fact, this is exactly what we would predict based on the observed kinetics for the oxide removal and growth. As the growth of the thick $PdO_x$ is slow, while its reduction is fast, just a little CO in the near-surface region will prohibit the growth of bulk oxides. In contrast, the √5 oxide growth proceeds fast and does not require high pressure of oxygen for seconds to form.

**Summary**. To summarize, we presented a method to generate event-averaged APXPS data from cyclic gas pulse experiments using a triggering signal obtained by image-recognition and originating from true changes on the surface. The event-averaging process removes time delays induced by history effects in the gas pulsing system or the sample due to e.g., segregation. This stands in contrast to what can be achieved if gas pulsing is used for triggering the measurement. The method is even suitable for non-periodic studies: for example, it becomes possible to study self-sustained oscillations that are non-periodic in nature[5,29]. Supplementary Fig. 3 shows how this can be achieved.

We have applied the method to the catalytic oxidation of CO over a Pd(100) surface at 3.2−100 mbar pressures. At 3.2 mbar we found that a √5 surface oxide acts as an oxygen reservoir that prohibits CO poisoning as it rapidly can be reduced. The √5 surface oxide remains on the surface as long as $O_2$ is found in the near-surface region and its rapid reduction and the transition to the CO-poisoned surface is first triggered once $O_2$ is nearly absent in the gas phase above the sample surface. The CO-poisoned surface remains inactive in low $O_2:CO$ ratios, but a key result of our study is that it turns active for a short time when the local $O_2:CO$ ratio becomes high enough to trigger the removal of the CO-poisoning effect by allowing chemisorption of oxygen in sufficient quantities to trigger $CO_2$ production. Once the CO mass transfer limit again is reached, CO desorbs fully and the oxygen reservoir in form of the √5 surface oxide reappears within 150 ms.

Similar experiments at 100 mbar pressure in a virtual cell with μm-dimensions allowed a much faster repetition rate and averaging over more than 500 events. The event-averaging method improved the data set to such a degree that the gas phase components, which were completely hidden in the noise in the single-event data, became fully analyzable. As in the 3.2 mbar experiments, we found a rapid removal of the $PdO_x$ phase under reaction-induced $O_2$-poor conditions. The re-oxidation required a high pressure of pure oxygen gas and was a significantly slower process.

In the work presented here, we reached time resolutions between 150 and 50 ms, limited by the software and framerate of the camera for the detector. Using faster software and cameras or a delay line detector it should be easy to increase the time resolution substantially. With such fast detectors, the time resolution for the image-recognition method presented here will practically be limited by the dwell time required to measure a spectral feature sufficiently intense to be used as a triggering signal in the raw data.

What makes the time-resolved method described here particular attractive is that it does not limit the finally achievable signal-to-noise ratio, which is proportional to the square root of events one averages over. Further, the regime where one approach or leaves the mass transfer limit can be studied with high time resolution. This is particular important since the phases responsible for driving the catalyst into or out of the mass transfer limit often exist for a short time. Importantly, it is possible to measure and compare catalyst activities in this regime in contrast to mass transport-limited reactions.

In our experiments, we used an oscillating gas composition to create forced surface phase oscillations. Many other sample environment parameters could be used in future experiments, such as the total gas pressure or the sample temperature. The method of image-recognition to generate a true surface triggering signal can be applied to any time-lapsed APXPS experiment[30] and even any surface science technique that uses a fast detector. The only requirement is that the time-resolved data contain features that can be identified by image recognition. The application to SXRD (see, e.g.,[28], and consider rapid gas pulsing and event-averaging applied to movies of diffraction patterns) or PLIF[27] and many other techniques should be straight forward. The method presented here thus opens entirely new avenues for time-resolved catalysis, as experiments with stroboscopic vision of surface changes in response to changes in the local sample environment now become possible.

## Methods

**Experiments at HIPPIE, MAX IV**. The data shown in Fig. 1, in Supplementary Figs. 4–7 and Supplementary Movies 1, 2, 4, 5, and 6 were all collected at the HIPPIE APXPS beamline during one single beamtime[22]. For these measurements, the Pd(100) single crystal was mounted on a transferrable 304 L stainless steel sample plate. An IR laser in closed-loop operation was used to heat the crystal and the temperature was monitored with a type K thermocouple spot-welded to the side of the crystals to ensure a precise temperature measurement. Before the experiment, the crystal was cleaned by 1 kV $Ar^+$ sputtering followed by annealing to 1070 K. The cleanness of the surfaces was confirmed by XPS survey scans.

Instrument oxygen (5.0 N) and CO (3.8 N) was used for the experiments. Commercial Pall gas cleaners (GLP2OXPVMM4 for $O_2$ and GLP2SIPVMM4 for CO) were used on both gas lines. This is particularly important for the CO line, as nickel carbonyls are known to form in the CO bottle. The gases were dosed with mass flow controllers (Brooks GF125) controlled by software written by MAX IV. The stated flow values in sccm units refer to standard conditions of 20 °C and 1 bar absolute pressure. The pressures stated in the paper were measured with a capacitance manometer mounted at the outlet of the cell. Gases were pre-mixed in the gas panel and dosed into the reaction cell through one common (~10 m long) inlet gas line. Due to the limited flow and ~ 1 l volume of the cell, this caused the $CO:O_2$ ratio in the cell to slowly rise (45 s) and fall (100 s). See Supplementary Fig. 1 for a simulation. The surface was exposed sequentially to 58 combinations of a 45 s long CO-rich ($CO:O_2$ mixing ratio 2.7:1 and 9.6 sccm flow) and 100 s long $O_2$-rich ($CO:O_2$ mixing ratio 2.7:1 and 9.6 sccm flow) pulses.

During the exposure, we continuously recorded the x-ray photoelectron (XP) spectra in the 25 × 60 μm² (horizontal × vertical) footprint of the beam with a high photon flux on the order of $2 \cdot 10^{12}$ photons/s. At such a high photon flux the inelastic scattering of photoelectrons can dissociate weakly adsorbed surface species such as $O_2$[31] or desorb adsorbed CO and thereby potentially change the kinetics of the surface restructuring. We judge, however, that beam-induced effects are unlikely in the present case since the elevated temperature ensures efficient thermal dissociation of $O_2$.

For all measurements at the HIPPIE beamline, the R4000-HIPP-3 electron analyzer was operated in fixed acquisition mode. This mode makes use of the energy dispersion of the analyzer, which allows fast acquisition of entire spectra at fixed analyzer voltages. C 1s and Pd $3d_{5/2}$ spectra were measured with a photon energy of 410 eV, while O 1s spectra were measured with a photon energy of 650 eV. A large pass energy of 200 eV was used both for O 1s and C 1s, while 100 eV was used for Pd $3d_{5/2}$ such that the entire binding energy range (approximately 10% of the pass energy) could be covered by the electron analyzer in fixed acquisition mode. The frequencies and corresponding time resolution of 148 ms stated in the paper were calibrated from the known pulsing period (i.e., from the known time difference between two CO pulses) since the actual framerate was lower than the expected 17 Hz for the camera in use in the experiments due to software issues.

**Experiments at POLARIS, DESY**. The POLARIS instrument equipped with a R4000-HiPP-2 electron analyzer, situated at the P22 beamline at Petra III synchrotron, was used to collect the data presented in Fig. 2 and Supplementary Fig. 8, as well as Supplementary Movies 3 and 7. This setup takes advantage of a double-cone for gas dosing, a small footprint of the beam (15 × 15 μm²), a small 30 μm aperture of the analyzer, and a very short sample-cone distance on the order of 30 μm, which leads to the creation of a *virtual cell*, in which the flow onto the sample creates a local high-pressure pillow with μm dimensions. A total flow of 1800 sccm was used and resulted in a local gas pressure of ~100 mbar in the virtual cell. The acceptance of the electron analyzer is matched to the beam footprint[24].

Gases of CO (purity 4.7), $O_2$ (purity 5.0), and He (purity 5.0) were used and further cleaned using gas purifiers obtained from SAES ($O_2$ gas, model Nr. 906; He gas, model Nr. 902). To avoid the formation of Ni carbonyls, a combination of Cu piping and a gas purifier for CO (PALL, GLP8SIPVMM4) was used. The cell pressure was measured indirectly from the pressure in the first differential pumping stage. Pressure calibration was performed in direct connection to the beamtime by measuring cell pressure as a function of the pressure in the first differential pumping stage for CO, $O_2$, and He, respectively with the sample fully retracted and therefore no creation of a virtual cell[24].

The pre-cleaned sample was transferred through air and introduced into the POLARIS instrument where it was treated by repeated cycles of oxidizing/reducing conditions under elevated pressure and temperature conditions. Judging from XPS, this procedure created a clean surface with a minor $SiO_2$ contamination. The Pd crystal was mounted on a stainless-steel sample holder and heated from the backside using a fully enclosed ceramic heater (model 101275-28 K, Heat Wave Labs). Type N thermocouples were mounted on the backside of the crystal and the temperature was stabilized using a remote controllable PID loop.

The data were collected at a grazing x-ray incidence angle of 0.7° at a photon energy of 4.6 keV using the Si (311) monochromator settings. In this combination, an effective probing depth of ca. 17 Å is expected based on the method from Jach et al.[32], wherein the penetration depth of the x-rays at and below the critical angle, as well as the escape of the electrons at those depths, is used to characterize the signal from the surface.

The photon energy band-width was ca. 130 meV. Using a slit of 0.8 mm and 200 eV pass energy, the combined experimental resolution was ca. 430 meV.

**Data analysis**. All data analysis was performed in Igor Pro 8 using purpose-written scripts. After event-averaging, a 5th order polynomial background subtraction was carried out for each spectrum by fitting the polynomial to the datapoints where no components were visible. To determine the datapoint range for fitting the background subtraction the sum spectrum of all time-resolved data was used (to ensure that the background subtraction did not remove any weak components). In this work, we did not correct for the different electron transmissions in the electron analyzer, which can affect the intensity. After background subtraction, all spectra were curve-fitted with asymmetric pseudo-Voigt functions using the build-in Voigt93 function described in Igor's technical notes. The asymmetric Voigt functions are obtained by adding two Voigt functions with different widths in the peak point. First width parameters (Lorentzian and Gaussian), asymmetry, binding energy, and intensity were adjusted and later curve-fitted manually for a few selected spectra along the time axis of each image plot. Subsequently, when fitting every spectra of the image-plot all parameters except the intensity parameter and a common work function shift parameter were fixed. These constraints made the curve-fitting procedure very robust. Finally, the quality of the curve-fitting was inspected by curve-fitting movies, which we also used to document our data analysis.

## Data availability

All datasets generated and analyzed during the current study are available from the corresponding author upon reasonable request, since private communication is necessary for the raw data analysis.

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

## Acknowledgements

We acknowledge MAX IV Laboratory for time on Beamline HIPPIE under proposal 20190578. Research conducted at MAX IV, a Swedish national user facility, is supported by the Swedish Research council under contract 2018-07152, the Swedish Governmental Agency for Innovation Systems under contract 2018-04969, and Formas under contract 2019-02496. We acknowledge DESY (Hamburg, Germany), a member of the Helmholtz Association HGF, for the provision of experimental facilities. Parts of this research were carried out at PETRA III and we would like to thank Christoph Schlueter for assistance in using beamline P22. Beamtime was allocated for proposal I-20190596 EC. The staff at the MAX IV Laboratory and DESY are gratefully acknowledged for support during measurements. We acknowledge Edvin Lundgren, Lund University, for enlightening discussions on the oxidation and reduction chemistry of Pd(100). Financial support from the Swedish Research Council, grant numbers 2017-04840, 2017-03871, and 2013-8823, Knut and Alice Wallenberg Foundation under Grant No 2016.0042, the Swedish Foundation for Strategic Research under project number ITM 17-0034, and the Research Council of Norway, project number 280903, Per Westlings foundation is gratefully acknowledged.

## Author contributions

J.K. has written the manuscript with M. Sc., A.N., P.A., A.S., J.S. and with input from all other authors. Experiments were conducted by T.G., V.B., M.D.S., G.D., C.G., S.Z., M. So., P.L., F.C., M. Sc., D.D., P.A. and A.S. J.K analyzed the data and developed the experimental idea.

## Funding

## Competing interests

The authors declare no competing interests.
