## [Peer Review File · Nature Communications]

Title: Stroboscopic operando spectroscopy of the dynamics in heterogeneous catalysis by event-averagingREVIEWER COMMENTS

Reviewer #1 (Remarks to the Author):

See attached PDF. 
Reviewer #2 (Remarks to the Author):

The authors report a novel “stroboscopic” method for studying catalytic reactions using time resolved XPS. The technique is creative and useful. They exposed a Pd(100) sample to periodic gas-pulses of different CO:O₂ composition while simultaneously acquiring XPS data, which probes both the surface and the gas-phase in a small volume in direct contact with the surface. They show that the key to obtaining time-resolved data is to identify an accurate “trigger” using image recognition algorithms so that many pulses can be reliably averaged and sensitivity obtained. This new method of studying dynamic phenomena in catalytic reactions has potential to be impactful. As such, the paper is generally suitable for publication in Nature Communications.

However, I find it difficult to identify the new insights about CO oxidation on Pd(100) that their study advances. I believe that there is new knowledge but the authors should clarify and better emphasize it. As an example, the main finding stated in the abstract is essentially that all surface phases (i.e., Pd metal, surface and bulk oxide) catalyze the CO oxidation reaction. This is not new knowledge. In my opinion, for the paper to be suitable for publication in Nat. Comm., the authors need to demonstrate that their stroboscopic method provides new understanding of CO oxidation on Pd(100) and clearly state what new knowledge has been obtained. Experimentally confirming earlier ideas may suffice such as demonstrating that the metal is active during the transition period when the CO coverage is high, for example. Other comments:

- 1) I suggest that the authors clarify in the paper the characteristic timescale that the reactant gases are switched in their experiments. Is this fast relative to the changes observed with XPS? This comment may be related to the next one.
- 2) A key to their method is the ability to simultaneously interrogate the temporal responses of the gas and surface compositions/chemical states that are stimulated by a forced time variation in the reactant composition. The exact time at which the reactant composition is changed seems unimportant. As such, I believe that relating the gas switching times to the times at which the surface and gas-phase change is not necessary. Assuming that my understanding is correct, I encourage the authors to discuss this in the paper – it was initially unclear to me.
- 3) Fig. 1E: Why does the CO partial pressure begin to decrease at 20 s? Is it caused by adsorption? This is difficult to understand because changes in other quantities (CO₂, CO(ad)) are not evident at 20 s?
- 4) The onset of CO₂ production at 47 s, a few seconds before the r5 oxide reappears, is convincing evidence that the metal surface becomes active and transitions into the so-called CO-uninhibited regime immediately before oxide formation. The authors may want to state that this transition seems to be

triggered by a sufficient increase in the gas-phase O₂:CO ratio, which presumably “lifts” the CO poisoning effect and allows oxygen to chemisorb on the metal in sufficient quantities to trigger CO₂ production. The coincidence of the increasing O₂:CO ratio and the onset of CO₂ production at this point is not clearly stated in the paper. This observation may represent a new finding about CO oxidation on Pd(100) as noted above.

5) I suggest stating how the 163 ms time for oxide formation is determined. It may not be clear to all readers.

6) Line 314: Typo – “high CO production” should state “high CO₂ production”

7) In Figure 2E, the second increase in CO₂ production rate is attributed to the gas mixture becoming stoichiometric. However, in Fig. 2E, the CO₂ partial pressure is the only quantity (among those shown) that changes between about 1 and 2 s. The CO:O₂ ratio appears to remain constant and the Pd 3p and O 1s signals also remain unchanged. Thus, the factors which trigger this second onset of CO₂ production are unclear in this case. This is different from Fig 1E where a steady decrease in CO:O₂ ratio is evident. What then is the trigger? Might there be changes in the state of the surface that are not resolved with XPS collected using 4.6 keV photons? The authors need to clarify this in the paper.

8) The authors conclude that the r5 oxide is “extremely resistant to reduction” by CO at 550 K. This wording could be interpreted to imply that the r5 oxide, being resistant to reduction, has a low activity for CO oxidation. I do not think this is necessarily what the authors intend to conclude, though it may be an appropriate conclusion. Notably, the thick PdOx phase undergoes very rapid reduction by CO. Do the authors think that these differences indicate that the thick PdOx phase is more reactive than the r5, at least for the different conditions studied? Such a conclusion would be consistent with the intrinsic reactivities of the r5 vs. thicker PdO(101) phases. However, is it instead that the longer time scale for reduction of the r5 vs. thick oxide is related only to the lower pressure used in the former, and that the observed rates are mass-transfer limited in both cases? They should be clearer and more explicit in their conclusions here, to the extent possible.

Reviewer #3 (Remarks to the Author):

I enjoyed reading this article. It is very well written and reports about the application of a stroboscopic, lock-in-like approach to study the state of an active catalyst surface and its response to – and interconnection with - the gas-phase composition above it.

For many years, catalysts and their active sites were described in a rather static picture. This work is very important also because it provides yet another clear demonstration of the complex dynamic interplay between reactive gas-phase and the actual working state of a catalyst.

The authors have selected CO oxidation over Pd(100), a very fundamental reaction, to demonstrate the potential and capabilities of combining a lock-in-like approach with time-resolved ambient pressure photoelectron spectroscopy. They demonstrate how the fundamental problem of a low signal-to-noise ratio in time-resolved APXPS can be overcome by event-averaging over many surface phase transitions. Well-visible shifts in the work function were used as lock-in-signal for the event-based integration. The obtained spectra show a good signal-to-noise ratio and are well suited for principle component analysis.

The amazing time resolution allows detailed investigation of the response of the systems to gas pulses. The interpretation of the results shows that all surface phases – the CO-covered Pd surface, a surface oxide and a thick PdOx phase – catalyse the CO oxidation reaction in dependence of the supply of gas phase reactants. These important mechanistic insights are in agreement with earlier findings, but here, as far as I can see, more complete, as the study clearly shows the interconnection between the working state of the catalyst and compositional variations in the gas-phase, since both are recorded simultaneously.

I cannot provide much constructive criticism, and would recommend to publish the manuscript basically as it is.

There are only two minor points that the authors could eventually consider in a revised version:

On page 2, in the discussion of imaging techniques that use strong probe signals, it is stated that: “However, these techniques do not provide access to activity and local gas composition and can become challenging for detailed surface information when the catalyst temperature is elevated.” As far as I am concerned, I would say that this is not strictly true: In the case of operando electron microscopy, it is possible to detect catalytic activity through compositional analysis of the gas-phase, either using electron energy-loss spectroscopy or on-line mass spectroscopy (although this bears risks associated to comparing a very local observation with integral spectroscopic data).

Then I miss a short discussion on the influence of the high photon flux: The influence of the photons or the emitted photoelectrons on the chemical activation of species, the surface coverage/surface chemical state should be estimated or addressed.

And finally, it would be nice (maybe I missed it) to know something about the beam diameter and associated lateral integration. That would be interesting especially in view of propagating reaction-fronts (chemical waves) as they are known to occur in many surface reactions.

REVIEWER COMMENTS

Reviewer #1 (Remarks to the Author):

In the manuscript “Stroboscopic operando spectroscopy of the dynamics in heterogeneous catalysis by event averaging”, Knudsen *et al.* present the instrumental developments in the gas delivery in APXPS combined with a new way to analyse the obtained data. Via event averaging, somewhat similar to averaging in pump-probe experiments, the noisy snapshot spectra taken with video rate are transformed into high-quality spectra, while maintaining a high time resolution. Generally, the work is convincing, well presented, and clearly written. This contribution fits in a trend of maturing of APXPS experiments, i.e., going towards more complex and powerful ways to control gas delivery. I believe this paper is interesting for researchers using APXPS setups at other synchrotrons or setups using lab-based X-ray sources. I think it is worth including more technical details and discuss the limitations of the presented technique.

Reply: We are very grateful for the careful reading of our manuscript and the very constructive critics that reviewer 1 given. We are also delighted that the reviewer finds our work convincing and interesting, in particular for scientists using APXPS. We want, however, to highlight that the method of finding a triggering signal in time-resolved 1D, 2D, 3D data that signals a structural change on the surface can be used for many other surface science techniques than APXPS. The methodology is, therefore, general in nature and we are convinced that the same methodology also will be very useful for time-resolved studies using for example SXR or PLIF (see also point 1 below).

1) In the summary, the authors mention that this method can be applied to non-periodic studies (line 338-339). I fail to understand how this method can be applied to non-periodic dynamics. The crux of the method is that averaging a periodic signal improves the signal-to noise ratio. Mentioning of self-sustained reaction oscillations in this regard is very interesting and shows exactly my problem understanding this. The reaction oscillations for CO oxidation on Pd(100) that we observed (<https://doi.org/10.1080/2055074X.2017.1280641>) had quite some variation in terms of period length. For the event averaging to work, one would have to compress/expand the time axis for individual events to ensure that each set of averaged spectra correspond to the same point in relative time. Therefore, applying this analysis to non-periodic signals is non-trivial and requires explanation.

Reply: We agree that analysis of non-periodic signals is non-trivial and requires a better explanation. To facilitate this we simulated non-periodic oscillations and made a new figure S3 for the SI that explains how such data can be event-averaged. We are happy to move the entire discussion of this into the paper, but as we did not studied self-sustained oscillations in the present work we felt that it is better positioned in the SI. In addition the paper is already at the length limit. In the revised version of the paper we refer to figure S3 when we mention our method also can be used for non-periodic studies on p. 10, l. l. 383-385.

2) The time resolution is impressive and will probably be good enough to study many of the slower reactions in catalysis (poisoning, roughening, dealloying, etc.) as the authors note: “millisecond to second time resolution is sufficient to follow structural surface changes of the catalyst, such as phase transitions, surface roughening and segregation.” (lines 44-45) However, the scientific example presented by the authors indicates that there is a need to go to a much higher time resolution; the most interesting aspect, the reduction and formation of the $\sqrt{5}$ oxide, is too fast to gain additional insight in. It would be a good thing if the authors discuss the current limitation in time resolution and strategies to improve this.

Reply: It is correct that we are unable to follow the gradual reduction and formation of the $\sqrt{5}$ oxide in a time-resolved manner. Nevertheless, we do believe that our work gives additional insights both about reduction and the growth of the $\sqrt{5}$ oxide. We show for example what triggers both reduction and oxidation by following the $\sqrt{5}$ oxide and the localized gas composition that the surface equilibrates towards. By doing this we show that, the rapid $\sqrt{5}$ oxide removal coincides with lack of oxygen supply,

while its appearance coincides with lack of CO supply. Furthermore, we show that the metallic and mainly CO covered surface becomes active first, reach the MTL, and subsequently the $\sqrt{5}$ oxide develops. So we do get new insights about the reduction and formation of the $\sqrt{5}$ oxide.

To follow the gradual reduction and formation of the $\sqrt{5}$ oxide in a time-resolved manner one would need higher time-resolution as pointed out by the reviewer. In the two examples presented in the article the time-resolution was limited by the software (HIPPIE) and the framerate of the camera (POLARIS) and we reached time resolutions of 150 and 50 ms, respectively. Faster software, cameras, or the use of a delay line detector (DLD) can increase the framerate substantially. 500 Hz cameras are for example commercially available and DLDs easily reach ns time resolution. With such fast detectors the time-resolution for the image-recognition method presented here will practically be limited by dwell time required to measure a spectral feature intense enough to be used as triggering signal in the raw data and how precise one can determine the absolute time of the triggering signal. In fact, we just been granted beamtime at a cell-based APXPS beamline equipped with a DLD detector and plan here to investigate the maximum time-resolution we can achieve with the image-recognition method. This beamtime is, however, first scheduled in the Autumn and we do not want to delay publication any further. We felt we already demonstrated the potential of the method as all reviewers also acknowledge and we also have clear scientific take home messages.

To go beyond the time resolution set by the dwell time required to measure a spectral feature intense enough to be used as triggering signal one needs to rely on external triggering signals and assume strictly periodic pulsing and response of the catalyst surface. Also this we studied with a dedicated puls-pump setup for another catalyst surface and one article is submitted where we demonstrate sub-ms time-resolution.

In the revised version, we added a short paragraph, which briefly discuss the current limiting factors for the time-resolution and strategies to improve it for the image-recognition method discussed here (l. 402-407, p. 11). We see the image recognition as the most novel part of the paper and also want to keep this the focus of the paper. Therefore, we do not discuss external triggering in the paper. External triggering will be discussed in our submitted article and in an extensive comparison of time-resolved APXPS experiments which will appear in an upcoming book chapter entitled "Time Resolved Ambient Pressure X-ray Photoelectron Spectroscopy".

3) The scientific example shows another limitation, which is the detection limit for measuring surface adsorbates: the catalyst's reactivity changes drastically, while the XPS signal is completely unchanged (Fig 1E, 40 to 50 s). This is a significant limitation. I would be interested in learning possible strategies to lower the detection limit, for example smaller pass energies, more event averaging, longer integration times, etc. To some extent, it may not possible to go both for video rate data acquisition while maintaining an acceptable low detection limit.

Reply: It is correct that the XPS gas phase signal of CO₂ increase substantially from 40 to 50 s in Fig. 1E, while the adsorbed CO signal remains constant. It nicely shows that even a Pd(100) surface mainly covered by CO is active for the CO oxidation reaction and is in our view not a limitation of the technique. In fact, this is an important finding as the CO-covered surface previously was believed to be completely inactive in much of the published literature on the Pd(100) system. See also our answers to the questions of reviewer 2.

To get an idea about the detection limit one can look at the signal to noise level of the adsorbed CO signal in figure 1E. This is in the range of 5:1, corresponding to a minimum detection limit of 20% of the CO saturation level. The reason for this relative large value is that we only event averaged over 58 very noisy spectra with a high background signal in the O 1s region originating from secondary electrons. For comparison the signal to noise level of the corresponding C 1s CO_{ads} signal is much better (see figure S7 D) and clearly above 10:1 meaning that less than 10% of the CO saturation level easily can be detected.

The signal to noise ratio of the data can in be reduced by event-averaging over more events, as it is proportional to \sqrt{N} , where N is the number of events averaged. One can easily increase the number of events one averages over by increasing the repetition frequency. We demonstrate how this can be done by using the POLARIS setup and an ultra-small cell. These results are discussed in figure 2. In the revised version of our paper we discuss the fact that the achievable signal-to-noise ratio or detection limit is unaffected by the method itself if enough events are averaged for APXPS or any other surface science techniques on p. 11, l. 408-413.

4) The ultimate time resolution of the surface dynamics interacting with the gas phase that can be achieved in the method presented here will depend on the detection mechanism (analyser settings, event averaging, etc.) *and* on the diffusion characteristics of the chamber/cell. The former is well described, but the latter is mostly missing. It would helpful to add this information, such as cell/chamber volume, refresh rate, pumping speed, and the delay between switching gases and new gas mixture arriving at the sample. It may be best to demonstrate the flow characteristics with inert gasses so that all changes can be ascribed to the flow dynamics and not to the catalyst changing the local gas phase.

Reply: We did measure on different pulses without an active catalyst present as the reviewer suggested and we plan to submit these data to a more technical journal once we they been fully analysed. To reserve some space for this upcoming publication we do not like to include these data in this publication. Instead, we simulated the gas flow in the cell based on the cell volume and flow rates in a new figure S1. One important point is, however, that the gas-phase that the surface responds to is setup by the combined effect of the gas dosing and the catalyst activity. A key advantage of APXPS is that the gas phase is measured locally.

Nevertheless, we agree with reviewer 1 that we should have included more experimental details such that readers better can judge the potential of the technique. This is now done in the method section.

5) Aligning the snapshot spectra using a signal in the spectra detected with imaging recognition is a clever trick, which the authors present as the main advantage of this technique. Although I don't disagree that this is helpful, it may simply not possible for all studies, e.g. the signal-to-noise ratio of individual spectra might be too low. So, it would be great to have the external trigger together with the internal trigger signal. The difficulties of an external trigger signal are briefly mentioned: "times delays induced by history effects in the gas pulsing system" (line 336-337). Further explanation or a reference would be nice to show the extent of this problem. I feel that there are many scientific/technical applications that are dependent on reliable pulsing/switching of gases (e.g. TAP, SSITKA, or GC), and that therefore the technical limitations have been resolved.

Reply: It is correct that TAP, SSITKA, and GC rely on reliable pulsing and the gas systems and reactors used for these techniques are therefore designed for generating this. For example, thin tubes, small reactor volumes, and short distances from the pulsing valves to the reaction cell are used in such setups. In contrast, reactors and gas systems for APXPS (and many other surface science) setups have been designed based on for example the requirement to perform an efficient bakeout of the cell and the gas system. For this reason, such setups use tubes with relatively large inner diameter, large cell volume, and mass flow controllers or pulsed valves placed far away from the reaction cell. This can lead to gas composition in the reactor volume that both changes slowly ("history effect in the gas pulsing system") and rapidly due to the pulsing. For this reason, it is doubtful that the 1st, 5th, 10th pulse is identical and one should in general be careful to event-average over the external pulsing signal in such setups. Rather, than rebuilding synchrotron based APXPS setups (and other complicated setups SXR, PLIF etc.), which for practical reasons often is impossible the methodology presented here offers a direct and easy way to implement event-averaged and time-resolved measurements. So indeed, one can build reliable pulsing/switching systems for gas pulses, but it is not always possible to interface this with surface sensitive setups (having other design requirements) often used by many different users.

We did not find any good reference for “history effects in the gas pulsing system” in the literature for TAP, SSITKA, or GC most likely as they rely and assume reproducible pulsing if event-averaging is used. For APXPS we did not find any peer-reviewed article either discussing this. We however, often observed history effects ourselves as we look for it and easy can detect it with our method (in contrast to methods that rely on it). Examples of this can be seen in figure 11 and 13 in the master thesis of Harald Wallander entitled “Observing phase changes in real time on ultrathin Iron oxide surfaces”, which can be downloaded from:

<https://lup.lub.lu.se/student-papers/search/publication/8987707>

In the revised version of our paper we explained the “history” effect of gas pulsing much more carefully on p. 4, l. 139-154. Furthermore, we explicitly point to figure S5 that show that our pulses do not are 100 % periodic.

Some small comments

a. It is unclear what “strong probe signals” and “weak ... probe signals” are (lines 47,48, and 54). Is this related to the ease with which artefacts (eg beam damage) are generated by a probe or maybe to how surface sensitive a probe is? Both TEM and STM are classified as being “strong probe”. I don’t think TEM and STM share any characteristic, except that electrons are in some way involved.

Reply: The intention with this formulation was to highlight that the cross section for creating a photoelectron is low. Using intense synchrotron light one typically only reach $\sim 1 \cdot 10^6$ electron per seconds on a MCP detector and only a fraction of these are core electrons. Therefore, the typically acquisition time for a photoelectron spectrum is at the order of several seconds to minutes to obtain a decent signal to noise ratio. In contrast, the convoluted morphology and electronic structure measured with scanning tunnelling microscope use a tunnelling current typically in the nA regime and equivalent to $\sim 6 \cdot 10^{15}$ electrons per second. For this reason, one can image many pixel points (typical values range from $64 \times 64 \approx 4 \cdot 10^3$ to $1024 \times 1024 \approx 10^6$) even with video rate frequencies. A similar argument can be applied for TEM measurements.

Nevertheless, we agree with the reviewer that the use of “strong” and “weak” probing signals are difficult to follow in the original version and it is not really needed. In the revised version, we therefore removed the discussion of strong and weak probe signals on p. 2.

b. It would help to add vertical dashed lines in Figs 1E and 2E to indicate the relative time at which spectra i and ii were taken.

Reply: This is a good idea since it helps the reader to quickly understand the figure. We added vertical lines as suggested to figure 1E and 2E.

c. What is happening with the Pd3p_{3/2} peak above dashed line ii in Fig 2B? It seems to grow a lot in intensity. Why is that? Is there a decrease in pressure?

Reply: The Pd 3p_{3/2} component increase due to the 1 s He pulse at each side of the 2 s CO rich pulse. In the revised version of the manuscript, we mentioned the reason the growth of the Pd 3p_{3/2} peak on p. 7, l. 211-213.

Very importantly, this comment made us realise an error in the analysis used for generating figure 2E. As we use He gas in this experiment one can only find the relative gas composition of CO₂, O₂, and CO, not the corresponding partial pressures. This does not change any of the conclusions of our paper, but we are extremely grateful that the comment by the reviewer made us capture this mistake. In the revised version of the manuscript we clearly stated that only relative gas pressures are obtained in figure 2E on p. 7, l. 230 – 234. Also figure 2E has been changed correspondingly.

d. “Regarding the 1 mbar experiment” (page 8). Shouldn’t this be 3 mbar?

Reply: We are grateful that the reviewer captured this mistake. This error has been corrected everywhere in the revised version.

e. “The reduction sets in first when the gas composition above the sample surface is dominated by CO₂ and the surface system is in the oxygen mass-transfer limit.” (page 8). Studying Fig. 1E, I reach a different conclusion: as soon as some of the CO is measurable, the intensity of the $\sqrt{5}$ oxide is decreasing (see the difference in slopes of the lines in the Fig below). This is when the gas phase is still dominated by O₂. To me it suggests that even the smallest amount of CO is enough so that CO oxidation competes with replenishment of the surface oxide, therefore an increasing CO pressure decreases the steady-state coverage of the surface oxide. I am assuming that the steep drop-off at t=0 s occurs because there is not enough O₂ in the gas phase left to render the surface oxide the most stable phase. I believe that this is a thermodynamic phase transition and not a kinetic transition as is proposed in lines 265-268.

Reply: The O 1s spectra were measured with a photon energy of 650 eV resulting in a kinetic energy of the photoelectron of around 120 eV for the two components assigned to the $\sqrt{5}$ oxide. At t = -40 s the gas composition in front consist of approximately 1 mbar CO₂ and 2.2 mbar O₂ referring to figure 1E, while the gas composition at t = 0 s consists of approximately 2.4 mbar CO₂ and 0.8 mbar O₂. The total cross section for O₂ and CO₂ for 120 eV electrons are around $8 \cdot 10^{-16} \text{ cm}^2$ and $1.2 \cdot 10^{-15} \text{ cm}^2$, respectively. Using these values the total cross section (neglecting the small amount of CO in the gas phase) can be calculated at t = -40 s and t = 0 s [DOI: 10.1140/epjd/e2011-20630-1]:

$$\sigma(-40 \text{ s}) = \frac{2.2 \text{ mbar}}{3.2 \text{ mbar}} \cdot 8 \cdot 10^{-16} \text{ cm}^2 + \frac{1.0 \text{ mbar}}{3.2 \text{ mbar}} \cdot 1.2 \cdot 10^{-15} \text{ cm}^2 = 9.3 \cdot 10^{-16} \text{ cm}^2$$

$$\sigma(0 \text{ s}) = \frac{0.8 \text{ mbar}}{3.2 \text{ mbar}} \cdot 8 \cdot 10^{-16} \text{ cm}^2 + \frac{2.4 \text{ mbar}}{3.2 \text{ mbar}} \cdot 1.2 \cdot 10^{-15} \text{ cm}^2 = 1.1 \cdot 10^{-15} \text{ cm}^2$$

The conclusion from this is that the total cross section increase by approximately 20% when the CO₂ concentration increase. This is mirrored by a 20% reduction of the $\sqrt{5}$ oxide which decrease from a mean value of 0.85 to 0.67 referring to figure 1E. So initially, we concluded that the apparent reduction of the $\sqrt{5}$ oxide is a cross section effect, which we did not wanted to discuss in detail in the a general paper like this.

Nevertheless, this very good comment of the reviewer made us revisit the data again and in fact, the Pd 3d_{5/2} data shown in figure S6 shows the same trend. Note that these data are normalised to the total Pd 3d_{5/2} intensity and changed cross sections in the gas phase should thus not cause any change to Pd spectra. Therefore, we tend to agree that slow reduction indeed is observed before one reach the oxygen mass transfer limit that then triggers rapid removal of the oxide.

In the revised version of the manuscript, we change the discussion accordingly on p. 8, l. 266-276.

f. “The reaction is catalysed by the PdO_x phase that is present at the surface at t = 0.0 s. The complete conversion of CO is accompanied by a reduction of the oxide, which is fully consumed within 250 ms.” (page 9) I don’t think these data directly show that PdO_x is a catalyst, which is suggested in the first part of this sentence. It could just as well be a stoichiometric reaction between CO and O in the PdO_x (as is suggested in the second part of the sentence).

Reply: We agree that the use of “catalysed” in this sentence is wrong as the oxide at the same time is consumed. In the revised version of the manuscript we changed the wording accordingly on p. 9-10, l. 340-351.

g. Regarding the data analysis (SI), I think it necessary to mention the order of the polynomial that was used in the background fitting, what the exact formula is for the asymmetric pseudo-Voigt used for fitting the peaks, and what parameters were free/fixed during fitting.

Reply: In the method section we added the order of the polynomial used for background subtraction, information about how one can find the exact formula of the numerical calculations of the Voigt functions we used in our work, and further details about how the curve-fitting were performed. In addition, we want to highlight, that we documented all curve-fitting extensively by making movies of the curve-fitting so every reader can inspect our curve-fitting.

Reviewer #2 (Remarks to the Author):

The authors report a novel “stroboscopic” method for studying catalytic reactions using time resolved XPS. The technique is creative and useful. They exposed a Pd(100) sample to periodic gas-pulses of different CO:O₂ composition while simultaneously acquiring XPS data, which probes both the surface and the gas-phase in a small volume in direct contact with the surface. They show that the key to obtaining time-resolved data is to identify an accurate “trigger” using image recognition algorithms so that many pulses can be reliably averaged and sensitivity obtained. This new method of studying dynamic phenomena in catalytic reactions has potential to be impactful. As such, the paper is generally suitable for publication in Nature Communications.

However, I find it difficult to identify the new insights about CO oxidation on Pd(100) that their study advances. I believe that there is new knowledge but the authors should clarify and better emphasize it. As an example, the main finding stated in the abstract is essentially that all surface phases (i.e., Pd metal, surface and bulk oxide) catalyze the CO oxidation reaction. This is not new knowledge. In my opinion, for the paper to be suitable for publication in Nat. Comm., the authors need to demonstrate that their stroboscopic method provides new understanding of CO oxidation on Pd(100) and clearly state what new knowledge has been obtained. Experimentally confirming earlier ideas may suffice such as demonstrating that the metal is active during the transition period when the CO coverage is high, for example. Other comments:

Reply: We are very grateful for the careful reading of our manuscript and the very constructive critics that reviewer 2 gave. We are also delighted that the reviewer find our work creative, useful, and similar to us also believe it will have impact. After reading the very thoughtful comments of reviewer 2 and scrutinizing our manuscript we also fully agree with reviewer 2 that we originally put too much focus on trying to explain the technique in a short and concise way and somehow forgot to highlight the important new insight that our study gives. In the revised version, we explicitly highlight our main

findings including: (i) That a transient metallic and predominantly CO covered surface turns highly active at 3.2 mbar total pressure for a few seconds once the O₂:CO ratio becomes high enough to lift the CO poisoning effect before mass transport limitations triggers formation of a $\sqrt{5}$ oxide. (ii) That the $\sqrt{5}$ surface oxide acts as an oxygen reservoir that prohibit CO poisoning as it rapidly can be reduced. (iii) That the rapid $\sqrt{5}$ oxide formation and removal correlates with the absence of CO and O₂, respectively, in the gas phase just above the sample.

Examples of this are found in the last part of the abstract (l. 24-27) and in the manuscript l. 102 – 106, l. 290-295, 334-339, and 386-395. In summary, we thus believe that comments by reviewer 2 really improved our paper substantially.

1) I suggest that the authors clarify in the paper the characteristic timescale that the reactant gases are switched in their experiments. Is this fast relative to the changes observed with XPS? This comment may be related to the next one.

Reply: This is a good comment and we agree that this is important to highlight that timescale of the switching in the cell is slow compared to the structural changes on the surface observed with APXPS. A comment about this has been added p. 4 l. 146 – 147 in the revised version and it is also more carefully explained in the method section on p. 12, l. 439 - 442. Furthermore, we simulated the time-evolution of the CO and O₂ partial pressures in the cell based on the cell volume and the known flow rates and a new figure and its corresponding discussion have been added to the SI as figure S1.

2) A key to their method is the ability to simultaneously interrogate the temporal responses of the gas and surface compositions/chemical states that are stimulated by a forced time variation in the reactant composition. The exact time at which the reactant composition is changed seems unimportant. As such, I believe that relating the gas switching times to the times at which the surface and gas-phase change is not necessary. Assuming that my understanding is correct, I encourage the authors to discuss this in the paper – it was initially unclear to me.

Reply: This is a very good comment and the reviewer's understanding is perfectly correct. This comment and discussion our results with other colleagues made us realize that it is crucial that the unconventional way of event-averaging should be explained better. Part of the problem is maybe that many scientists knows very well how a traditional pump-probe experiment is performed, but our approach is different from this with clear advantages, but of course also with some limitations.

In the revised version we added a longer paragraph on p. 4, l. 135 - 176, that discuss our methodology in the context of a traditional pump-probe experiments.

3) Fig. 1E: Why does the CO partial pressure begin to decrease at 20 s? Is it caused by adsorption? This is difficult to understand because changes in other quantities (CO₂, CO(ad)) are not evident at 20 s?

Reply: The dosing of the CO-rich gas composition has a duration of 45 s. Inspection of figure 1 of the main manuscript and figure S7 shows that the arrival of the CO-pulse in the cell happens at -25 s and it is thus expected that the CO pressure should reach its maximum value after 45 s of dosing at t = 20 s in agreement with figure 1 and S7.

In the revised version of the manuscript we clarified why the CO partial pressure starts to decrease after t = 20 s on p. 8, l. 287-289.

4) The onset of CO₂ production at 47 s, a few seconds before the r5 oxide reappears, is convincing evidence that the metal surface becomes active and transitions into the so-called CO-uninhibited regime immediately before oxide formation. The authors may want to state that this transition seems to be triggered by a sufficient increase in the gas-phase O₂:CO ratio, which presumably “lifts” the CO poisoning effect and allows oxygen to chemisorb on the metal in sufficient quantities to trigger CO₂ production. The coincidence of the increasing O₂:CO ratio and the onset of CO₂ production at this point is not clearly stated in the paper. This observation may represent a new finding about CO oxidation on Pd(100) as noted above.

Reply: This is a very good comment and we implemented it in the last paragraph on p. 8, l. 290-295. In addition we added a short paragraph on p. 9, l. 334 - 339 to highlight that this is key finding of our paper.

5) I suggest stating how the 163 ms time for oxide formation is determined. It may not be clear to all readers.

Reply: We fully understand why the reviewer might find the reported $\sqrt{5}$ to CO covered surface switching time of 163 ms unjustified, as we reported a sampling frequency corresponding to a time resolution of 148 ms. This was unfortunately a mistake caused by a slight change in the time-calibration. Unfortunately, we overlooked to update the value of 163 ms to 148 ms in the old version of the manuscript.

In the revised version of the manuscript and the SI we corrected the old value of 163 ms to the correct one of 148 ms and clarified that the transition both from the oxygen – to CO covered surface and vice versa happens faster than time resolution dictated by our sampling frequency. (p. 8, l. 281 – 283 and p. 9, l. 307 - 309).

6) Line 314: Typo – “high CO production” should state “high CO₂ production”

Reply: We are grateful for the careful reading of our manuscript and for highlighting this confusing typo. We corrected this typo in the revised version (p. 10, l. 347-348).

7) In Figure 2E, the second increase in CO₂ production rate is attributed to the gas mixture becoming stoichiometric. However, in Fig. 2E, the CO₂ partial pressure is the only quantity (among those shown) that changes between about 1 and 2 s. The CO:O₂ ratio appears to remain constant and the Pd 3p and O 1s signals also remain unchanged. Thus, the factors which trigger this second onset of CO₂ production are unclear in this case. This is different from Fig 1E where a steady decrease in CO:O₂ ratio is evident. What then is the trigger? Might there be changes in the state of the surface that are not resolved with XPS collected using 4.6 keV photons? The authors need to clarify this in the paper.

Reply: Also this is a very good comment, which made us realise that the original version was unclear in the discussion of figure 2. Within the entire time domain shown in figure 2E the CO oxidation reaction is mass transfer limited. Clear evidence for this is the fact that we never observe significant quantities of CO and O₂ at the same time considering the error bars of the data points. First the mass transport of CO to the surface in excess O₂ is limited (-2 s < t < 0 s), then the mass transport of O₂ to the surface is limited (0 s < t < 2 s), and finally the mass transport of CO to the surface is limited again (2 s < t).

This situation is thus very different from figure 1. In the revised version of the paper we clearly explain this in a new paragraph on p. 10, l. 352-365.

8) The authors conclude that the r5 oxide is “extremely resistant to reduction” by CO at 550 K. This wording could be interpreted to imply that the r5 oxide, being resistant to reduction, has a low activity for CO oxidation. I do not think this is necessarily what the authors intend to conclude, though it may be an appropriate conclusion.

Reply: We agree that the wording “The $\sqrt{5}$ surface oxide turns out to be extremely resistant to reduction...” suggests that the oxide is passive and a spectator phase with a low activity for the CO oxidation reaction. Such a picture is in our view not correct as the surface once covered fully by the $\sqrt{5}$ surface oxide also oxidize CO to CO₂. To be more explicit in our conclusions and to ensure that the readers interpret our wording correctly we rewritten to: “These observations show that the $\sqrt{5}$ oxide converts all CO to CO₂ as long as sufficient O₂ is available in the gas phase to replenish the oxide.” p. 8, l. 283-285.

Notably, the thick PdOx phase undergoes very rapid reduction by CO. Do the authors think that these differences indicate that the thick PdOx phase is more reactive than the r5, at least for the different conditions studied? Such a conclusion would be consistent with the intrinsic reactivities of the r5 vs. thicker PdO(101) phases. However, is it instead that the longer time scale for reduction of the r5 vs. thick oxide is related only to the lower pressure used in the former, and that the observed rates are mass-transfer limited in both cases? They should be clearer and more explicit in their conclusions here, to the extent possible.

Reply: See also the reply to question number 7. Considering the rapid reduction of $\sqrt{5}$ and the PdOx phase both transitions happens when CO starts to reach the near surface area and most likely both transitions are mass transfer limited. Therefore, we do not think it is possible to say anything about the reactivity of the $\sqrt{5}$ and versus the PdOx phases. The main difference is instead the oxidation rate. The $\sqrt{5}$ oxide is formed rapidly once the partially CO covered surface driven the reaction into the MTL. In contrast, the PdOx phase takes seconds to develop as seen from figure 2E.

In the revised version we clearly state that the reaction always is mass transfer limited in figure 2 and it therefore is impossible to compare intrinsic reactivities of $\sqrt{5}$ and PdOx (p. 10, l. 352-365).

Reviewer #3 (Remarks to the Author):

I enjoyed reading this article. It is very well written and reports about the application of a stroboscopic, lock-in-like approach to study the state of an active catalyst surface and its response to – and interconnection with - the gas-phase composition above it.

For many years, catalysts and their active sites were described in a rather static picture. This work is very important also because it provides yet another clear demonstration of the complex dynamic interplay between reactive gas-phase and the actual working state of a catalyst. The authors have selected CO oxidation over Pd(100), a very fundamental reaction, to demonstrate the potential and capabilities of combining a lock-in-like approach with time-resolved ambient pressure photoelectron spectroscopy. They demonstrate how the fundamental problem of a low signal-to-noise ratio in time-resolved APXPS can be overcome by event-averaging over many surface phase transitions. Well-visible shifts in the work function were used as lock-in-signal for the event-

based integration. The obtained spectra show a good signal-to-noise ratio and are well suited for principle component analysis. The amazing time resolution allows detailed investigation of the response of the systems to gas pulses.

The interpretation of the results shows that all surface phases – the CO-covered Pd surface, a surface oxide and a thick PdOx phase – catalyse the CO oxidation reaction in dependence of the supply of gas phase reactants. These important mechanistic insights are in agreement with earlier findings, but here, as far as I can see, more complete, as the study clearly shows the interconnection between the working state of the catalyst and compositional variations in the gas-phase, since both are recorded simultaneously.

I cannot provide much constructive criticism, and would recommend to publish the manuscript basically as it is.

Reply: We are delighted that the reviewer enjoyed reading our manuscript and also find our work very important. Even-though the reviewer did not believe she/he gave much constructive critics we actually believe the comments she/he gave are very important to address. Also these comments been used to improved our manuscript and we are very grateful for this.

There are only two minor points that the authors could eventually consider in a revised version: On page 2, in the discussion of imaging techniques that use strong probe signals, it is stated that: “However, these techniques do not provide access to activity and local gas composition and can become challenging for detailed surface information when the catalyst temperature is elevated.” As far as I am concerned, I would say that this is not strictly true: In the case of operando electron microscopy, it is possible to detect catalytic activity through compositional analysis of the gas-phase, either using electron energy-loss spectroscopy or on-line mass spectroscopy (although this bears risks associated to comparing a very local observation with integral spectroscopic data).

Reply: This is a good comment. We overlooked that energy-loss spectroscopy can be used to measure the gas composition locally in the case of operando TEM. In the revised version of the article, we now mention that energy-loss spectroscopy can be used to measure gas composition locally (p. 2, l. 51-55) and included one recent reference for illustration of this. However, this technique has limited flexibility for the sample and its environment.

For the use of on-line mass spectroscopy, it (in our view) does not probe the gas composition locally in the same way as APXPS and energy-loss spectroscopy applied to operando TEM experiments and we therefore did not include this in the discussion in the revised version.

Then I miss a short discussion on the influence of the high photon flux: The influence of the photons or the emitted photoelectrons on the chemical activation of species, the surface coverage/surface chemical state should be estimated or addressed.

Reply: We agree that the high photon flux or rather the large amount of secondary electrons created are important to address. A particular elegant feature of the experiment is, however, that the photon flux is constant in the experiment, while only the gas composition changes, which in turn changes the surface phase. Therefore, it is unlikely that the photon flux causes the oscillations. Further evidence for this statement comes from the fact that, we never observed any beam damage effects in the (many) CO oxidation studies performed on Pd(100) at the HIPPIE beamline. In the revised version of the article, we included a short paragraph on p. 12, l. 443-448 where beam damage effects are shortly discussed.

And finally, it would be nice (maybe I missed it) to know something about the beam diameter and associated lateral integration. That would be interesting especially in view of propagating reaction-fronts (chemical waves) as they are known to occur in many surface reactions.

Reply: We agree that this is important information and therefore added beam diameters and acceptance of the electron analysers used at setups at HIPPIE and POLARIS and in the revised manuscript, we briefly mention this in the method section (p. 12, l. 443-448 and p. 12 l. 459-466).

Propagating reaction fronts as Ertl observed at UHV conditions would definitely be interesting to study in the mbar regime, but it is probably best studied with ambient pressure PEEM or LEEM, which recently became available. Nevertheless, APXPS experiments could also be used as the HIPP-3 electron analyser used at the HIPPIE beamline also has a spatial-resolved mode with resolution below 15 μm , but only along a stripe. Using this mode, one would be able to answer questions such as whether or not the phase change happens simultaneously at different points of the sample. Future experiments, along this direction is definitely interesting and something we will pursue, but we felt that discussion about potential future experiments is beyond the scope of the present article.

REVIEWERS' COMMENTS

Reviewer #1 (Remarks to the Author):

After reading the replies by the authors to the barrage of comments, I believe that the manuscript has been improved substantially. Therefore, I think that this manuscript is a great contribution to Nature Communications and I recommend publication in the current form.